# The Effects of Methylprednisolone and Hyaluronic Acid on the Endometrium in Experimentally Induced Asherman Syndrome Rat Models: A Prospective Laboratory Study

**DOI:** 10.3390/medicina61030482

**Published:** 2025-03-10

**Authors:** Mehmet Genco, Merve Genco, Fisun Vural, Nermin Koç

**Affiliations:** 1Department of Obstetrics and Gynecology, Haydarpaşa Numune Training and Research Hospital, Istanbul 34668, Turkey; mervedogan93@windowslive.com (M.G.); fisunvural@yahoo.com.tr (F.V.); 2Department of Pathology, Haydarpaşa Numune Training and Research Hospital, Istanbul 34668, Turkey; nerminkoc@yahoo.com

**Keywords:** Asherman Syndrome, methylprednisolone, hyaluronic acid, histopathological score, VEGF, prospective laboratory study

## Abstract

*Background and Objectives*: The current study was designed as a prospective laboratory investigation to evaluate the histopathological effects and VEGF (vascular endothelial growth factor) expression in uterine tissue following treatment with a combination of methylprednisolone and hyaluronic acid in a rat model of experimentally induced Asherman Syndrome. *Materials and Methods*: Twenty-six female Sprague-Dawley rats were used. Trichloroacetic acid was applied to the right uterine horns of all the groups to induce adhesion formation. First, we induced the Asherman model in two rats (Group 1). The remaining rats were divided into the following three groups: Group 2 received intrauterine hyaluronic acid treatment, Group 3 received oral methylprednisolone treatment, and Group 4 received both treatments. Inflammation, gland count, and fibrosis levels were assessed histopathologically. VEGF levels were analyzed immunohistochemically. *Results*: Hyaluronic acid treatment increased the uterine lumen diameter and vascularization. Methylprednisolone treatment increased the gland count and uterine wall thickness while decreasing the inflammation and fibrosis scores. Combined treatment provided a statistically significant advantage over single treatments. In particular, the combined treatment group exhibited significantly lower fibrosis (*p* = 0.184) and inflammation scores (*p* = 0.071), as well as higher gland counts (*p* = 0.849) and VEGF expression (*p* = 0.114), compared to the groups receiving only methylprednisolone or hyaluronic acid. These differences indicate that the synergistic effect of the two agents results in more effective endometrial healing than when either treatment is applied alone. *Conclusions*: Methylprednisolone treatment significantly prevented adhesion formation and reduced the inflammation and fibrosis scores compared to hyaluronic acid treatment alone. The combined treatment adds to the effects of the hyaluronic acid treatment alone and provides better healing.

## 1. Introduction

Intrauterine adhesions (IUA), also known as Asherman Syndrome (AS), involve the formation of scar tissue within the uterine cavity. This condition was first described by Heinrich Fritsch in 1894, and the term “Asherman Syndrome” was later coined by Josef Asherman in 1948 [1,2]. Since then, numerous cases and various treatment protocols have been reported. Despite advancements in hysteroscopic adhesiolysis, adhesion recurrence remains a major challenge, leading to persistent infertility and reproductive complications. Therefore, novel therapeutic approaches are needed to improve endometrial healing and reduce fibrosis after intrauterine injury.

Patients with Asherman Syndrome exhibit a range of symptoms, including infertility, hypomenorrhea, amenorrhea, menstrual irregularities, pregnancy loss, obstetric complications, and placental invasion anomalies [3]. However, the diagnosis and optimal treatment strategy for AS remain debated. The incidence of Asherman Syndrome varies depending on the studied population, being 4.6% among infertile women, 37.6% following abortions, and 40% after recurrent curettage [4]. The primary causes of IUA include trauma and infections affecting the basal layer of the endometrium. Risk factors include curettage, myomectomy, Mullerian anomaly surgeries, uterine embolization, B-lynch sutures, and genital tuberculosis [5,6,7,8].

While hysteroscopy is the gold standard for both diagnosing and treating IUA, preventing post-surgical adhesion recurrence remains a significant challenge [9]. Various strategies, including intrauterine devices, Foley catheters, gel barriers, estrogen therapy, and regenerative medicine approaches (e.g., platelet-rich plasma, stem cells), have been explored to promote endometrial regeneration and prevent re-adhesion [7,9,10,11,12,13]. However, these strategies have shown inconsistent efficacy, and no universally accepted treatment protocol has been established.

This study aims to address this gap by evaluating a combination of methylprednisolone (MP) and hyaluronic acid (HA) as a novel therapeutic approach to prevent intrauterine adhesion formation. Previous studies have investigated HA as a mechanical barrier and MP as an anti-inflammatory and antifibrotic agent in other fibrotic conditions, such as idiopathic pulmonary fibrosis and keloid scars. However, no prior study has systematically evaluated the combined effect of HA and MP in AS treatment. This research aims to determine whether the dual approach using both MP (suppressing inflammation and fibroblast proliferation) and HA (providing mechanical protection and promoting tissue repair) offers superior therapeutic outcomes compared to single-agent treatments.

To bridge the gap between the clinical presentation of Asherman Syndrome and our experimental approach, we developed a rat model that simulates the condition. Trichloroacetic acid (TCA) was chosen as the method to induce AS because it closely mimics the inflammatory and fibrotic responses observed in human IUA. In TCA-induced inflammation, tissue damage triggers a cascade of immune responses, beginning with the migration and accumulation of polymorphonuclear leukocytes (PMNs) in the acute phase and transitioning to macrophages in the chronic phase. This inflammatory progression mirrors the pathways observed in AS, where the retained products of conception act as irritants, leading to macrophage activity and fibroblast activation, ultimately promoting fibrosis.

Steroids such as methylprednisolone (MP) are known to suppress inflammation, reduce vascular permeability, and inhibit fibroblast proliferation. They achieve this by inhibiting chemotactic factors and cytokine production, thereby preventing excessive fibrin and collagen buildup that contributes to adhesion formation [14,15]. While MP has been widely used in inflammatory and fibrotic disorders, its application in intrauterine adhesion prevention remains largely unexplored. On the other hand, HA is commonly used as an adhesion-preventing agent in gynecological surgeries, as it provides a physical barrier, promotes fibrin degradation, and enhances mesothelial cell regeneration [16]. Given these complementary mechanisms of action, this study hypothesizes that combining MP and HA could provide an optimal balance between reducing inflammation and promoting endometrial regeneration.

Endometrial vascularization plays a critical role in regeneration and fertility, and vascular endothelial growth factor (VEGF) is a key regulator of this process [17]. Impaired VEGF expression has been associated with poor endometrial receptivity, infertility, and increased fibrosis in patients with Asherman Syndrome. Steroids, while reducing fibrosis, may also suppress VEGF expression due to their anti-inflammatory properties, whereas HA has been shown to promote angiogenesis and wound healing. By evaluating the immunohistochemical expression of VEGF, we aim to assess whether MP’s antifibrotic effects negatively impact vascularization and whether HA can counteract this effect, providing insight into the potential synergistic role of these two agents in improving endometrial repair.

Given the limited research on combination therapies for AS, this study was designed to investigate and compare the efficacy of hyaluronic acid, methylprednisolone, and their combination in a TCA-induced Asherman Syndrome rat model. We hypothesize that the combination therapy will lead to reduced fibrosis, improved histopathological healing, and enhanced VEGF expression compared to single treatments. If successful, this approach could offer a novel strategy for improving post-operative outcomes in patients undergoing treatment for intrauterine adhesions.

## 2. Materials and Methods

In this study, the experimental induction of Asherman Syndrome (AS) and the subsequent treatment protocols were conducted according to previously established standardized methods. The surgical principles described by Jing et al. [18] and Hunter et al. [19] were followed for the creation of the AS model, while the trichloroacetic acid (TCA) application adhered to the protocol outlined by Kılıç et al. [20].

### 2.1. Animal and Ethical Approval

A total of 26 non-pregnant female Sprague-Dawley rats (200–250 g) were used. Ethical approval was obtained from the Yeditepe University Animal Experiments Ethics Committee (HADYEK: 2022/01-2), and the study was conducted in accordance with international guidelines (Guide for the Care and Use of Laboratory Animals). Before conducting the study, we performed a power analysis (with α = 0.05, 80% power) to determine the minimum sample size needed to detect moderate-to-large effect sizes in our primary endpoints (e.g., fibrosis score, gland count, inflammation levels). Using pilot data from a preliminary trial (and/or relevant literature on TCA-induced Asherman Syndrome models), we estimated that at least six rats per group would be sufficient to achieve adequate statistical power. We consequently chose an initial sample size of eight rats per group, anticipating possible surgical or postoperative losses.

### 2.2. Study Design and Estrous Cycle Assessment

All animals underwent a 1-week acclimatization period. During this time, vaginal cytology was performed to determine the estrous cycle phase, ensuring that surgical and experimental interventions were timed consistently across all animals.

### 2.3. Establishment of the Asherman Syndrome Model

The AS model was created following the principles reported by Jing et al. [18] and Hunter et al. [19]. For anesthesia, 6 mg/kg Xylazine-HCl (Rompun^®^, Bayer, Istanbul, Turkey) and 85 mg/kg Ketamine-HCl (Keta-control^®^, Mefar Ilac, Istanbul, Turkey) were administered intramuscularly. After shaving and disinfecting the abdominal area, a 2 cm laparotomy incision was made. The right uterine horn was gently clamped at the top and cervical levels. Following the Kılıç et al. [20] protocol, 0.1 mL of TCA (IL 33^®^, Istanbul Ilac Sanayi ve Ticaret AS, Istanbul, Turkey) was injected into the right uterine horn with an insulin syringe. The right uterine horn was chosen because, as reported in previous experimental models, it is commonly preferred due to easier surgical access and procedural consistency. A sponge was used to prevent acid leakage, and the abdomen was irrigated with sterile saline before closure. The uterus was allowed to develop adhesions over 14 days. Initially, AS formation was confirmed histologically in 2 rats, and subsequently, the same procedure was applied to the remaining 24 rats (Figure 1). Figure 1 presents a comparative histopathological evaluation of the experimental groups against the control. Figure 1A shows a normal left uterine horn (H and E, ×100). Figure 1B illustrates the Asherman model in the right uterine horn, characterized by a decreased number of glands, reduced uterine wall thickness, and lumen diameter (H and E, ×100). Figure 1C highlights the Asherman model in the right uterine horn with markedly increased fibrosis (Masson Trichrome, ×100), emphasizing the severity of fibrotic changes compared to untreated or treated groups. After the postoperative loss of 4 rats, 22 rats remained that were randomly assigned to experimental groups. Four rats were lost during the postoperative period, primarily due to adverse effects related to TCA application. The severe local inflammatory reactions induced by TCA, which sometimes led to complications such as infection and shock, were responsible for these losses. Despite careful monitoring and supportive care, the unpredictable toxicity associated with TCA resulted in the death of four animals.

### 2.4. Groups and Treatment Protocols

Group 1 (Control, n = 2): Induction of AS model, no further treatment (Figure 1). This group was utilized solely to confirm the induction of Asherman Syndrome using TCA, and in subsequent groups, it was accepted that Asherman Syndrome was induced with TC.Group 2 (Hyaluronic Acid, n = 7): Fourteen days (approximately 3 estrous cycles) post AS induction, a second laparotomy was performed, and 0.01 mL of low molecular weight hyaluronic acid was injected into the right uterine horn.Group 3 (Methylprednisolone, n = 6): Fourteen days post AS induction, 1 mg/kg p.o. methylprednisolone was administered daily for 2 weeks. Kazemi K et al. (2024) explored corticosteroid use in adhesion prevention models and concluded that 1 mg/kg MP provided significant anti-inflammatory and antifibrotic benefits with minimal systemic side effects. Given these findings, we selected 1 mg/kg as a well-balanced dose expected to exert therapeutic effects on intrauterine adhesion formation while maintaining endometrial regeneration capacity [21].Group 4 (Combined Treatment, n = 7): Fourteen days post AS induction, a second laparotomy was performed for intrauterine injection of 0.01 mL hyaluronic acid, and 1 mg/kg p.o. methylprednisolone was given daily for 2 weeks.

All procedures were performed under similar conditions and within the same timeframe to maintain experimental parallelism and ensure that all animals are exposed to comparable environmental, physiological, and hormonal conditions.

The experimental duration was set at 15 days post-treatment to evaluate the early histopathological and immunohistochemical effects of methylprednisolone and hyaluronic acid on endometrial healing. This timeframe was selected based on prior studies examining short-term inflammatory and regenerative responses in intrauterine adhesion models. Given that the rat estrous cycle is approximately 4–5 days, a 15-day period corresponds to three to four full cycles, which allows for the assessment of initial endometrial remodeling and VEGF expression.

While this study provides valuable insights into early-stage endometrial recovery, it may not fully capture long-term tissue remodeling, fibrosis regression, or functional reproductive outcomes. A longer follow-up period (e.g., 4–6 weeks) could provide additional information on whether the improvements in gland count, uterine lumen structure, and VEGF expression are sustained over time. However, prolonged observation periods in animal models introduce potential confounding variables, such as spontaneous tissue remodeling and hormonal cycle variability. Future studies should incorporate extended follow-up durations to evaluate the persistence of histopathological improvements and their impact on reproductive function.

By referencing these established protocols [19,20,22,23] and clearly detailing the experimental steps, the methodology is both standardized and transparent, increasing the study’s reproducibility, reliability, and clarity for the readers.

### 2.5. Pathological Evaluation

All histological evaluations were performed in a blinded manner, with the pathologist unaware of the treatment group assignments.

Hematoxylin and Eosin staining: The shape of the endometrial epithelium, gland structure and number, uterine wall diameters, uterine cavity lumen, and inflammation level were evaluated with semi-quantitative inflammation scoring using Hematoxylin and Eosin staining.

Masson’s trichrome staining: The severity and extent of fibrosis in the uterus and the fibrosis area ratio (fibrosis area/analyzed uterine area × 100) were evaluated. Fibrosis areas were determined as a percentage. Grade 0: no fibrosis; Grade 1: minimal fibrous tissue increase; Grade 2: irregular fibrous tissue increase; Grade 3: concentric fibrosis, hyalinization was present.

Immunohistochemical staining was performed using ThermoScientific Immunostaining for anti-VEGF (PB9071 Boster Bio, Pleasanton, CA, USA). In this immunohistochemistry protocol, VEGF expression was evaluated on 2–3 sections per sample. After deparaffinization and rehydration, antigen retrieval was performed in citrate buffer (pH 6.0). Endogenous peroxidase activity was blocked using 3% hydrogen peroxide. The sections were then incubated with a primary anti-VEGF antibody (diluted per the manufacturer’s guidelines), followed by a peroxidase-conjugated secondary antibody. Chromogenic detection was achieved using 3,3′-Diaminobenzidine (DAB). DAB serves as a substrate for peroxidase; when oxidized in the presence of hydrogen peroxide, it produces an insoluble brown precipitate at the site of the antigen, clearly marking VEGF-positive areas. Finally, sections were counterstained with hematoxylin, and VEGF staining intensity along with the percentage of positive cells was semi-quantitatively scored on a 0–3 scale as follows: VEGF staining in <20% of cells—IHC score: 1; VEGF staining in 20–60% of cells—IHC score: 2; and VEGF staining in >60% of cells—IHC score: 3 [22].

Inter-observer reliability for histopathological scoring was positively assessed. In addition to the primary evaluation performed by a blinded pathologist, a subset of slides was independently scored by a second pathologist. The agreement between the two observers was high, confirming the consistency and reliability of our histopathological assessments.

### 2.6. Statistical Analysis

This study used SPSS 24.0 IBM statistical package program. The data were analyzed at 95% confidence intervals. Descriptive statistics are given as mean, standard deviation, median, minimum, and maximum values. We used Shapiro–Wilk’s test for normality testing. The Mann–Whitney U test and the Kruskal–Wallis test were used to analyze pairwise group comparisons that did not show a normal distribution. Analyses with *p* < 0.05 were considered statistically significant. To address the issue of multiple comparisons across four experimental groups (control, hyaluronic acid, methylprednisolone, and combination therapy), we applied the Kruskal–Wallis test for overall group differences, followed by Dunn’s post hoc procedure with Bonferroni correction. This approach helps control the family-wise error rate when comparing multiple treatment groups, reducing the risk of Type I errors. For pairwise comparisons (e.g., right vs. left horn within a single group), we used Mann–Whitney U tests if the data were non-parametric. Additionally, effect sizes were calculated to enhance the interpretation of statistical significance. Eta-squared (η^2^) was used as a measure of effect size for Kruskal–Wallis tests, while rank-biserial correlation (r) was applied for Mann–Whitney U comparisons.

## 3. Results

Adhesion formation was confirmed histologically after using trichloroacetic acid (Figure 1). The uterine wall thickness of the right uterine horn was significantly reduced in Group 1 (*p*: 0.004) and Group 2 (*p*: 0.035). Groups 3 and 4 had similar uterine wall thickness in the right and left horns. Table 1 shows the uterine wall thickness of the right and left horns. In Group 3, methylprednisolone was administered orally, which means that its systemic effects were distributed throughout the body and, consequently, affected both uterine horns. As a result, the anti-inflammatory and antifibrotic actions of methylprednisolone were not limited to the TCA-damaged horn but were also observed in the untreated horn, leading to similar uterine wall thickness measurements. This outcome was expected given the systemic nature of oral steroid administration.

Figure 2 shows a comparison between the lumen diameters of Group 1, Group 2, Group 3, and Group 4 (×100, Hematoxylin and Eosin).

Table 2 shows a comparison between the right and left uterine cavities. Group 1 had a significantly decreased lumen diameter in the right uterine horn compared to the left (*p*: 0.04). The uterine lumen diameters of the right and left horns were similar in Group 2, Group 3, and Group 4 (*p* > 0.05).

Group 1 (*p*: 0.029) and Group 2 (*p*: 0.039) had a decreased glandular count in the right horn compared to the left. Group 3 and Group 4 had similar glandular counts in the right and left uterine cavities (*p*: 0.077 and 0.847, respectively).

This indicates that the systemic effect of methylprednisolone (Group 3) and the combination therapy (Group 4) resulted in a balanced and homogeneous regeneration of endometrial glands between both horns. The *p*-value in Group 3 (0.077) was borderline; although a slight difference was observed—likely due to the sample size or individual variations—it did not reach statistical significance. In Group 4, the *p*-value (0.847) reflects an almost entirely balancing effect. Table 3 details these findings, demonstrating the impact of the treatment methods on endometrial glandular regeneration.

The right uterine cavities of Group 1 and Group 2 had increased inflammation compared to the left. Group 3 and Group 4 had similar inflammation levels in the right and left uterine cavities. Group 1 and Group 2 had significantly increased fibrosis in the right uterine cavity (*p*: 0.007 and 0.024, respectively). Group 3 and Group 4 had similar fibrosis in the right and left uterine cavities. Table 4 demonstrates a comparison between the inflammation levels and fibrosis in the right and left uterine cavities.

Group 1 and Group 3 had significantly decreased VEGF staining in the right uterine cavity (*p*: 0.011). Group 2 and Group 4 had similar VEGF staining levels in the right and left uterine cavities. Table 5 shows the immunohistochemical results of the VEGF levels.

The groups with treated right uterine horns (Groups 2, 3, and 4) were compared to the group with a mere induction of the Asherman Syndrome (Group 1) with respect to their lumen diameter, uterine wall thickness, gland count, inflammation (VEGF), and fibrosis (MT) levels using the Kruskal–Wallis test (Table 6). According to the test results, a statistically significant difference was observed in inflammation levels and lumen diameter between the treated groups and the untreated group with induced Asherman Syndrome (*p* < 0.05).

Post-hoc analyses following the Kruskal–Wallis test indicated that this significant difference was associated with the hyaluronic acid treatment. Hyaluronic acid treatment significantly increased uterine lumen diameter and reduced inflammation levels in Asherman Syndrome. These findings suggest that hyaluronic acid has beneficial effects in the treatment of Asherman Syndrome, particularly in preserving the uterine lumen diameter and reducing inflammation.

In addition to the *p*-values, the effect sizes were calculated to provide a more comprehensive interpretation of statistical significance. For Kruskal–Wallis tests, Eta-squared (η^2^) was used as a measure of the effect size, indicating medium to large effects across variables (e.g., Lumen Diameter η^2^ = 0.41, Fibrosis Score η^2^ = 0.34). Similarly, for Mann–Whitney U comparisons, Rank-Biserial Correlation (r) was applied, showing large effects in several key parameters (e.g., Fibrosis Score r = 0.71, Gland Count r = 0.67). These findings confirm that the observed differences are not only statistically significant but also clinically meaningful in terms of reducing fibrosis and enhancing endometrial regeneration.

## 4. Discussion

This study investigated the role of intrauterine hyaluronic acid and oral steroid usage in the treatment of Asherman Syndrome in an experimental rat model. Hyaluronic acid treatment facilitated uterine cavity expansion, while combined hyaluronic acid and steroid treatment resulted in increased uterine cavity expansion and VEGF levels, along with a significant reduction in fibrosis and inflammation. The results of this study indicated that both methylprednisolone and hyaluronic acid treatments significantly improved uterine healing in a rat model of Asherman Syndrome.

Steroids are used in treating many entities, such as reducing keloid scars, idiopathic pulmonary fibrosis, and preventing fibrosis in autoimmune hepatitis. In particular, methylprednisolone may improve postoperative tissue healing by reducing adhesion formation. However, it is not used in gynecology, especially to prevent intrauterine adhesion formation. This study investigated the effects of methylprednisolone use on the uterine cavity in Asherman Syndrome. Methylprednisolone treatment alone increased the uterine wall thickness, lumen diameter, gland count, and VEGF levels while reducing fibrosis and inflammation scores. Hyaluronic acid treatment primarily increased the lumen diameter and VEGF levels.

Previous studies have demonstrated the individual efficacy of MP and HA in reducing fibrotic adhesion formation. However, no study has systematically evaluated the combination of these agents in intrauterine adhesion prevention. Our results suggest that MP alone reduces excessive fibroblast activity and inflammation, while HA primarily serves as a mechanical barrier and enhances angiogenesis. Compared to previous studies, which mainly focused on the HA’s role in adhesion prevention, our findings highlight that the MP + HA combination therapy provides superior histopathological improvements, including reduced fibrosis, increased gland count, and enhanced endometrial vascularization. These findings emphasize the potential advantages of using HA not only as a passive barrier but also as a bioactive agent that supports the regenerative capacity of MP-treated tissue.

Asherman Syndrome presents a wide spectrum of clinical presentations, such as menstrual irregularities, oligo-hypomenorrhea, infertility, or recurrent pregnancy losses, making it a challenging condition for clinicians to treat. There is still no consensus on the optimal method to prevent Asherman Syndrome after intrauterine surgeries. Currently, there are many alternative approaches for the treatment of Asherman Syndrome. A commonly used method has been dilation and curettage. However, this blind procedure has a low treatment success rate and increases the risk of uterine perforation [6]. Today, hysteroscopy has replaced these methods. Hysteroscopy not only confirms the diagnosis of intrauterine adhesions but also provides a means of treatment. However, complications such as recurrent adhesion formation, uterine perforation, hemorrhage, and infection can occur during hysteroscopy [6,9,24,25]. The increasing rates of cesarean sections and intrauterine surgeries have made Asherman Syndrome an increasing problem, necessitating the search for new treatments.

The synergistic effects of methylprednisolone and hyaluronic acid can be attributed to their complementary mechanisms of action. Corticosteroids such as MP exert their anti-inflammatory and antifibrotic effects by downregulating TGF-β1 expression, inhibiting fibroblast activation, and reducing collagen deposition. However, long-term steroid use may impair tissue healing by suppressing angiogenesis and cellular proliferation. In contrast, HA acts as a regenerative agent by promoting extracellular matrix remodeling, stimulating mesothelial cell proliferation, and increasing VEGF-mediated angiogenesis [26,27]. Thus, while MP suppresses excessive inflammation and fibrosis, HA counterbalances the potential negative effects of steroids by supporting tissue repair and vascularization. This dual-action approach not only prevents intrauterine adhesion formation but also enhances endometrial regeneration, making it a promising therapeutic strategy. 

In a study by Kilic et al., an experimental Asherman model was created in two rats by injecting trichloroacetic acid into the uterine horns. Intrauterine adhesions were observed 2 weeks later using light microscopy [20]. Similarly, we used trichloroacetic acid to induce Asherman Syndrome and histopathologically confirmed it as an effective method.

In a double-blind randomized clinical trial by Tafti et al., 65 women who underwent uterine septum resection were divided into two groups. One group received 1 cc of hyaluronic acid gel immediately after the septal resection, while the control group received 1 cc of normal saline solution as a placebo injected into the uterine cavity. Two months later, the presence of intrauterine adhesions was examined by hysteroscopy in both groups. The study found that hyaluronic acid treatment significantly reduced the risk of developing Asherman Syndrome in women with endometrial damage after septal resection surgery [28].

Given that both MP and HA are already used clinically in different contexts (MP in inflammatory conditions and HA in adhesion prevention), a logical next step would be a phase I clinical trial assessing the safety and efficacy of intrauterine MP + HA application in women at risk of IUA. Such studies should evaluate not only adhesion recurrence rates but also endometrial receptivity and fertility outcomes over an extended follow-up period. This would provide the necessary clinical data to support the transition of this therapy from preclinical models to real-world clinical use. 

One major limitation of our experimental model is that the animal studies may not fully translate to human pathophysiology. Although the rat uterus shares certain features with the human endometrium, differences in estrous cycle length, immune responses, and healing mechanisms may affect how reliably these findings can be extrapolated. In particular, the regenerative capacity of the human endometrium is regulated by a complex menstrual cycle that differs significantly from the rat’s estrous cycle, which may limit the direct applicability of our results to human clinical scenarios.

A second key limitation is the relatively short follow-up period (15 days). Even though this timespan approximates 3–4 months in human physiology when adjusted for the rat’s 4–5-day estrous cycle, it still may not capture long-term outcomes, such as fertility success or sustained endometrial health. Longer-term studies would be necessary to determine whether fibrosis regresses over time or if additional interventions are required to maintain endometrial integrity.

A third important limitation is that while we investigated the therapeutic benefits of intrauterine methylprednisolone administration, potential adverse effects were not assessed. Corticosteroids are known to influence local immune responses, and prolonged exposure may lead to delayed wound healing, increased infection risk, or endometrial thinning. Further studies are needed to evaluate the safety profile of intrauterine methylprednisolone application, particularly its impact on long-term endometrial function and reproductive outcomes. 

Additionally, interspecies differences must be considered when interpreting these results. While the rat model effectively mimics the fibrotic response seen in intrauterine adhesions, the structural and functional differences between the rat and human endometrium may impact translational relevance. Specifically, the rat uterus lacks a true menstrual cycle, relying instead on an estrous cycle with distinct hormonal regulation. Future studies should incorporate primate or human ex vivo models to better assess the applicability of these findings to clinical settings.

Lastly, the small sample size reduced the statistical power to detect subtle differences between groups. Although the median values were comparable across different treatment conditions, some parameters did not reach statistical significance, which may have been influenced by sample size limitations. Future research with larger cohorts will be necessary to validate and strengthen these findings.

Future randomized controlled trials (RCTs) should assess not only the histological improvements in adhesion prevention but also functional reproductive outcomes, including pregnancy rates and implantation success. If confirmed in clinical settings, the MP + HA combination therapy could represent a novel approach for improving fertility outcomes in patients at risk of intrauterine adhesion recurrence.

## 5. Conclusions

Our results clearly demonstrate that the combined treatment with hyaluronic acid and steroids significantly improved the histopathological outcomes in our Asherman Syndrome model, suggesting potential clinical benefits. However, while these findings are promising, caution is required before translating them into human clinical practice. Specifically, further comprehensive preclinical and clinical studies are needed to clarify the safety, efficacy, and long-term outcomes of the combined therapy. Such studies are essential to confirm its potential to prevent fibrosis and improve fertility in patients with intrauterine adhesions.

## Figures and Tables

**Figure 1 medicina-61-00482-f001:**
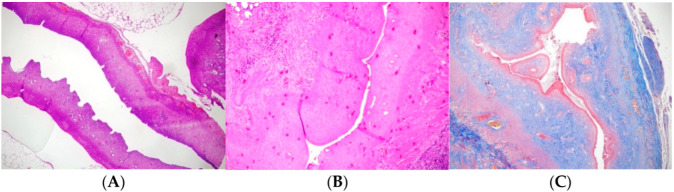
(**A**) Normal left uterine horn (×100, Hematoxylin and Eosin staining). (**B**) Asherman model in the right uterine horn (×100, Hematoxylin and Eosin staining). (**C**) Asherman model in the right uterine horn (×100, Mason Trichrome staining).

**Figure 2 medicina-61-00482-f002:**
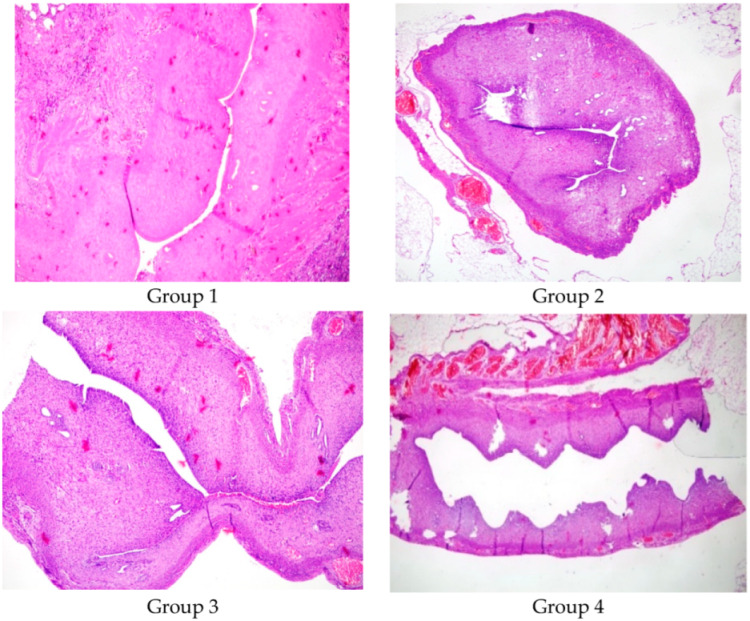
A comparison between the lumen diameter of Group 1, Group 2, Group 3, and Group 4 (×100, Hematoxylin and Eosin).

**Table 1 medicina-61-00482-t001:** The comparison between the uterine wall thickness of right and left uterine horns.

Uterine Wall Thickness (Micrometer)			
	Mean ± SD	Median (Min–Max)	*p* Value
Group 1 Right Left	892.5 ± 668.2	892.5 (420–1365)	0.004
	1879.5 ± 311.8	1879.5 (1659–2100)	
Group 2 Right Left	1041.4 ± 560.4	980 (430–1800)	0.035
	1915.3 ± 767	1870 (1100–3100)	
Group 3 Right Left	1715.9 ± 775	1229 (790–2000)	0.200
	1932.6 ± 768.4	1987 (680–4070)	
Group 4 Right Left	1270.7 ± 455.4	1877 (549–2875)	0.655
	2154 ± 1251.4	1975 (970–3350)	

**Table 2 medicina-61-00482-t002:** A comparison between the lumen diameter of right and left uterine cavities.

Uterine Cavity Lumen Diameters (Micrometer)			
	**Mean ± SD**	Median (Min–Max)	*p* Value
Group 1 Right Left	198.5 ± 16.3	198.5 (187–210)	0.004
	810 ± 381.8	810 (540–1080)	
Group 2 Right Left	500.4 ± 350.7	410 (188–1210)	0.064
	908.6 ± 373.3	980 (370–1490)	
Group 3 Right Left	536.3 ± 318.1	505 (369–1080)	0.055
	601.4 ± 291.3	1015 (436–1540)	
Group 4 Right Left	567 ± 260.9	405 (196–1005)	0.848
	1026.8 ± 382.4	549 (194–1054)	

**Table 3 medicina-61-00482-t003:** A comparison between the uterine gland counts from right and left uterine cavities.

Uterine Gland Counts			
	**Mean ± SD**	Median (Min–Max)	*p* Value
Group 1 Right Left	7 ± 2.8	7 (5–9)	0.029
	12.5 ± 5	12.5 (9–16)	
Group 2 Right Left	6.3 ± 2.2	6 (3–9)	0.039
	10.1 ± 3.9	9 (7–18)	
Group 3 Right Left	10.4 ± 5.4	7.5 (4–13)	0.077
	10.7 ± 4.4	11 (7–16)	
Group 4 Right Left	7.8 ± 3.1	10 (4–19)	0.849
	11.5 ± 3.5	10 (5–18)	

**Table 4 medicina-61-00482-t004:** A comparison between the inflammation levels in right and left uterine cavities.

Inflammation Levels				Fibrosis		
	**Mean ± SD**	Median (Min–Max)	*p* Value	Mean ± SD	Median (Min–Max)	*p* Value
Group 1 Right Left	2.5 ± 0.7	2.5 (2–3)	0.050	2.5 ± 0.7	2.5 (2–3)	0.007
	1.5 ± 0.7	1.5 (1–2)		1.5 ± 0.7	1.5 (1–2)	
Group 2 Right Left	1.6 ± 0.8	1 (1–3)	0.019	2.1 ± 0.7	2 (1–3)	0.024
	0.6 ± 0.5	1 (0–1)		1.1 ± 0.7	1 (0–2)	
Group 3 Right Left	0.7 ± 0.8	0.5 (0–2)	0.665	1.7 ± 0.8	1.5 (1–3)	0.101
	0.8 ± 0.8	1 (0–2)		0.8 ± 0.8	1 (0–2)	
Group 4 Right Left	1.3 ± 0.8	1 (0–2)	0.071	1.7 ± 0.8	2 (1–3)	0.184
	0.6 ± 0.5	1 (0–2)		1.1 ± 0.7	1 (0–2)	

**Table 5 medicina-61-00482-t005:** A comparison between the VEGF staining in right and left uterine cavities.

VEGF Staining Intensity			
	Mean ± SD	Median (Min–Max)	*p* Value
Group1 Right Left	1.5 ± 0.7	1.5 (1–2)	0.011
	2.5 ± 0.7	2.5 (2–3)	
Group 2 Right Left	1.9 ± 0.7	2 (1–3)	0.207
	1.4 ± 0.8	1 (1–3)	
Group 3 Right Left	1.3 ± 0.5	1 (1–2)	0.041
	2.3 ± 0.8	2.5 (1–3)	
Group 4 Right Left	1.6 ± 0.5	2 (1–2)	0.114
	2.1 ± 0.7	2 (1–3)	

**Table 6 medicina-61-00482-t006:** Comparison between right uterine horns of Group 1 (Asherman Induced) and Groups 2, 3, and 4.

	Uterine Cavity Lumen Diameters *p* Value	Uterine Wall Thickness *p* Value	Uterine Gland Counts *p* Value	Inflammation Levels *p* Value	VEGF Staining Intensity *p* Value	Fibrosis *p* Value
Right horn	0.010	0.107	0.321	0.009	0.483	0.162

## Data Availability

The datasets generated and/or analyzed during this study are available upon request. The materials used in the study are also accessible upon request for research purposes.

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
