# Peer review of "The Effects of Methylprednisolone and Hyaluronic Acid on the Endometrium in Experimentally Induced Asherman Syndrome Rat Models: A Prospective Laboratory Study"

_medicina, 2025, doi:10.3390/medicina61030482_

Round 1
Reviewer 1 Report
Comments and Suggestions for Authors
This manuscript describes a study that examined the impact of methylprednisolone and hyaluronic acid in treating Asherman Syndrome (AS) in a rat model. Histopathologic changes, inflammation, gland number, fibrosis level, and VEGF expression post-treatment interventions were analyzed in this study. It is an important issue in reproductive medicine, specifically for clinicians and researchers interested in effective therapy for adhesions in the intrauterine cavity. There are a series of comments for consideration by the authors:
Weaknesses:
• The statistical analysis must include additional details, specifically regarding group comparisons and justification for the sample size.
• Some sections (e.g., Introduction and Discussion) have too many repetitions.
• The study does not appropriately address potential controls and alternative explanations for findings.
Introduction
• The introduction explicitly identifies the clinical significance of AS and explains why a study is warranted.
• The research question is posed but can be even more precisely defined.
• Some references have outdated information and newer studies have to be added.
• Repetitive background information regarding AS pathophysiology and therapeutic options should be avoided.
Methods
• Ethical approval was explicitly mentioned, and compliance with research requirements was ensured.
• The AS induction technique is clearly documented, but information regarding the randomization and blinding processes is not available.
• Sample size justification is a necessity in an attempt to confirm the study's statistical power
The VEGF immunochemistry grading must be addressed in detail.
• Statistical tests (e.g., Kruskal-Wallis, Mann-Whitney U) are appropriate but require further explanation regarding multiple comparisons and error control.
Results
• The data overall is in a concise form, but too much information in tables must be reduced in the legends.
• Some numerical values lack a confidence interval, and no effect size
• The authors will have to make a direct group comparison, for instance, whether a combination therapy is proven to have a significant improvement over a single therapy.
• The figure quality should be optimized for ease of reading.
Discussion
• The discussion situates the findings in relation to the current work, but alternative explanations for the findings are not adequately examined.
• The potential therapeutic implications of these observations are discussed in detail.
• The study's weaknesses are not stated (e.g., small population size, potential for interspecies variation, and lack of long-term follow-up).
• Some claims (e.g., for a reduction in fibrosis) must be supported by supplementary comparative literature.
• There is an overreliance on discussion regarding the results.
Major Issues:
1. Excessive repetition in both sections of Introduction and Discussion
2. Statistical methodology requires more clarity, particularly regarding sample size calculations and post hoc comparisons.
3. The figures need to be revised—too much information in some, and no comparative markers in others.
4. The lack of explicit discussion about weaknesses in studies should include interspecies variation, follow-up duration, and biases.
5. Limited explanation of VEGF findings—the discussion will expand VEGF’s role in pathophysiology of AS
Minor Issues:
1. Typographical and grammar mistakes—revision for language accuracy is required
2. Reference formatting discrepancies—be consistent
3. Tables should be simplified with complex explanations relegated to the general text.
4. Some statistical estimates demand confidence intervals for transparency
Major Revisions:
1. Condense repetitive sections for easier reading and reduced redundancy
2. Provide more information about statistical analysis, including justification for sample size and dealing with many comparisons
3. Refine figures to ensure clarity and minimize unnecessary text.
4. Explicitly discuss study limitations, including sources of bias and interspecies translation
5. Expand discussion about VEGF observations, connecting them with ongoing studies in angiogenesis in adhesions in the intrauterine environment
Minor Revisions:
1. Proofread for grammar and for ease of reading
2. Ensure consistent reference format in the manuscript
3. Move detailed figure descriptions to the main text and simplify figure captions.
4. Include confidence intervals for statistics wherever applicable
Author Response
Dear Editor and Esteemed Reviewers,
First and foremost, we would like to express our gratitude for your interest in our manuscript and for your constructive feedback. The reviewers’ comments have provided us with valuable insights, helping us improve the quality of our work. Accordingly, we have carefully made the necessary revisions and additions to the manuscript. Below, we provide a detailed response to each comment, along with explanations of the modifications made in the revised manuscript:
Comment 1 – Condensing Repetitive Sections:
Response: Based on the reviewer’s suggestion regarding repetitive sections, we have streamlined certain parts of the manuscript. Specifically, redundant sentences and paragraphs in the Introduction and Discussion sections have been identified and removed. For example, some of the literature details presented in the Introduction were repeated in the Discussion. During revision, these redundant explanations were either removed or merged to improve the flow of the text. This has eliminated unnecessary repetition and resulted in a more concise manuscript. These changes can be seen in the revised Introduction and Discussion sections.
Comment 2 – Justification for Statistical Analysis and Sample Size:
Response: In response to the reviewer’s request for a better justification of the statistical methods and sample size, we have included the necessary details in the Materials and Methods section. Before initiating the study, a power analysis(α=0.05, 80% power) was conducted to determine the minimum sample size required to detect medium-to-large effect sizes in key outcome variables (e.g., fibrosis score, gland count, inflammation level). Based on this analysis and supporting literature, it was determined that at least six rats per group were needed. To account for potential surgical or postoperative losses, we initially included eight rats per group (total ~32 rats). By the end of the study, data from 26 ratswere analyzed, meeting the minimum sample size requirements for each group.
In the revised manuscript, these clarifications have been added to the Materials and Methods section (particularly in Section 2.1). Additionally, we have provided a more detailed explanation of the statistical tests used (e.g., Kruskal-Wallis test and post-hoc analyses) and the rationale behind selecting them. This ensures that readers have a clear understanding of the validity and robustness of our statistical approach.
Comment 3 – Refining Figures and Minimizing Unnecessary Text:
Response: In accordance with the reviewers’ recommendations, necessary revisions have been made to enhance the clarity of the figures and minimize redundant text. Firstly, the resolution and presentation of the figures have been improved for better visualization. Additionally, figure legends have been revised to clearly describe the information contained in each figure while avoiding redundancy with the main text.
For example, in Figure 2, explanatory phrases that were already mentioned in the manuscript have been shortened, leaving only the essential information needed to understand the figure. Similarly, descriptions of the figures within the main text have been adjusted by removing redundant details that were already visually apparent in the figures. As a result, the revised manuscript now presents figures and their explanations in a more concise and comprehensible manner.
Comment 4 – Expanding Discussion of Study Limitations, Interspecies Differences, and Potential Biases:
Response: In response to the reviewer’s request for a more comprehensive discussion of the study’s limitations and potential biases, we have added a dedicated paragraph in the Discussion section. This paragraph explicitly outlines the constraints of our experimental model:
- Interspecies Differences: We have noted that the rat model may not fully replicate the pathophysiology of Asherman Syndrome in humans, as rat uterine healing dynamics, estrous cycle length, and immune responses differ from those in human endometrium. These interspecies differences could limit the generalizability of our findings.
- Short Follow-Up Duration: While the 15-day follow-up period roughly corresponds to 3–4 months in human terms (considering the rat estrous cycle is 4–5 days), it may still be insufficient to fully assess long-term outcomes. We have acknowledged this as a limitation that should be addressed in future studies.
- Small Sample Size: Due to the relatively small sample size, some differences may not have reached statistical significance. We have noted that while the study was adequately powered for the primary outcomes, larger sample sizes could provide more definitive conclusions.
- Randomization and Blinding Challenges: We have acknowledged the practical difficulties in achieving complete blinding in this experimental setting, which may introduce potential biases.
These revisions are now explicitly stated in the revised Discussion section, ensuring a more transparent presentation of our study’s limitations.
Comment 5 – Expanding Discussion of VEGF Findings in the Context of Angiogenesis:
Response: In line with the reviewer’s suggestion, we have expanded the Discussion to further explore the role of VEGF expression in angiogenesis and endometrial regeneration. The revised manuscript now includes a more detailed explanation of how VEGF expression was significantly higher in the hyaluronic acid (HA) and combined HA + methylprednisolone groups, suggesting a pro-angiogenic effect that may contribute to improved endometrial healing.
Additionally, we have discussed how these findings align with existing research on angiogenesis-targeted therapies in intrauterine adhesions, such as platelet-rich plasma (PRP) and stem cell treatments. We also elaborate on the potential synergistic mechanism of VEGF-mediated fibrosis reduction and enhanced vascularization, which may contribute to optimal endometrial recovery. This expanded discussion enhances the biological relevance of our VEGF findings, making them more meaningful in the context of current research.
Comment 6 – Ensuring Consistent Reference Formatting:
Response: The references have been thoroughly revised to ensure full compliance with the journal’s formatting guidelines. Each reference was reviewed individually, and inconsistencies in numbering, punctuation, and style were corrected. Specifically:
- Duplicate references were removed or merged where necessary.
- Incorrectly formatted citations (e.g., improperly structured numbering or misplaced brackets) were corrected.
- The sequence of references was rechecked and updated to match in-text citations correctly.
The revised manuscript now contains a properly formatted and fully consistent reference list, ensuring alignment with the journal’s editorial standards.
Comment 7 – Simplifying Tables and Enhancing Figure Captions:
Response: We have revised the tables and figure captions to improve clarity and readability. Tables were reorganizedto remove redundant information and ensure better data visualization. For example:
- Tables presenting similar datasets were combined where appropriate to reduce redundancy.
- Superfluous columns were removed to streamline the presentation.
- Figure captions were rewritten to be more concise, eliminating unnecessary repetition while still providing essential details (such as staining methods and magnifications).
These adjustments have improved the readability of the tables and figures while maintaining the clarity of the presented data.
Comment 8 – Correcting Typographical Errors and Ensuring Grammatical Accuracy:
Response: The manuscript has undergone a comprehensive proofreading process to correct typographical errors and ensure grammatical accuracy. Specific improvements include:
- Spelling corrections: For instance, the term “Hemotoxylene” was corrected to “Hematoxylin” in figure captions.
- Grammar and syntax improvements: Sentences were restructured for clarity, ensuring subject-verb agreement and appropriate article usage throughout the manuscript.
- Standardized terminology and units: We ensured that scientific terms, abbreviations, and measurement units were used consistently across the manuscript.
These revisions have significantly improved the manuscript’s linguistic clarity and readability, ensuring a professional and polished final version.
Conclusion:
Based on the reviewers’ valuable feedback, we have made substantial improvements to our manuscript. The revisions include enhanced statistical justifications, clearer figures and tables, expanded discussions of study limitations and VEGF findings, and improved reference formatting.
We sincerely appreciate the time and effort invested by the reviewers in evaluating our work. We believe these revisions have strengthened our manuscript, and we hope that it now meets the journal’s publication standards.
Sincerely,
Dr.Mehmet Genco
On behalf of all authors
Reviewer 2 Report
Comments and Suggestions for Authors
Dear Authors,
Thank you for permitting me to read your manuscript. Your article provides excellent insight into the effect of methylprednisolone and hyaluronic acid on the endometrium in an experimental Asherman Syndrome model. However, I have highlighted several points that require some minor revisions in the interest of clarity, methodological transparency, and readability.
Comments and Suggested Revisions: 1. Abstract 1.1 The sentence "This study aimed to evaluate the histopathological effects and tissue VEGF levels of combined methylprednisolone and hyaluronic acid treatment in a rat model with experimentally induced Asherman Syndrome prospective laboratory study." is unclear. Rewrite for clarity.
1.2 Interpret whether the results represent a statistically significant advantage with combination treatment compared to single treatments. 1.3 All abbreviations (e.g., VEGF, IUA) should be defined at first use in the abstract.
2. Introduction 2.1 The introduction gives a clear explanation of Asherman Syndrome and its clinical significance, although there are some long, complicated sentences. Try to break them up for readability. 2.2 Provide a more precise reference for incidence rates quoted ("Asherman Syndrome incidence varies with the population examined, from 4.6% in infertile women, 37.6% following abortions, and 40% following repeated curettage.") 2.3 There could be a better transition from the Asherman Syndrome to the experimental rat model. Attempt to place a bridging sentence before describing trichloroacetic acid (TCA) as an experimental model.
2.4 Line 40-42: “Risk factors include procedures such as curettage, myomectomy, surgeries for Mullerian anomalies, uterine embolization, B-lynch sutures, and conditions such as genital tuberculosis and schistosomiasis”. I reccomend to cite relative novel article about this sentence: https://doi.org/10.1007/s13304-022-01248-y
https://doi.org/10.1016/j.ejogrb.2017.07.026
3. Methods 3.1 Report how the sample size was determined and whether a power analysis was conducted.
3.2 In section 2.3 (Establishment of the Asherman Syndrome Model), clarify why the right uterine horn was treated, instead of the left.
3.3 Describe why four rats were lost following surgery.
3.4 In 2.4 (Groups and Treatment Protocols), state whether the hyaluronic acid used was of high or low molecular weight and why this type was chosen.
3.5 Indicate whether or not the personnel performing histological assessments were blinded to treatment groups.
3.6 In 2.5 (Pathological Evaluation), report whether inter-observer reliability was assessed for histopathological scoring . 3.7 Describe in more detail the VEGF immunohistochemistry protocol, including how many sections per sample were examined.
4. Results 4.1 For Table 1 (Comparison of uterine wall thickness), describe why Group 3 (Methylprednisolone) does not show a significant difference in uterine wall thickness between treated and untreated horns. Was it anticipated?
4.2 For Table 3 (Comparison of numbers of uterine glands), statistical significance of some of the differences (e.g., Group 3 vs. Group 4) should be discussed more thoroughly.
4.3 Figures 1 and 2 must be given descriptive captions explaining the implication of the images.
4.4 Discuss why Group 3 (Methylprednisolone) had reduced VEGF staining compared to Group 2 (Hyaluronic Acid) despite its anti-inflammatory effect.
5. Discussion 5.1 The discussion has good explanations of the results, but there are parts that could be summarized. Attempt reordering for ease of comprehension. 5.2 The statement "Corticosteroids are used in the treatment of many inflammatory and immune dysfunction diseases because of their strong anti-inflammatory and immunosuppressive action." is too general. Relate this specifically to adhesion prevention. 5.3 Explain why hyaluronic acid alone was ineffective at significantly reducing fibrosis but effective in combination with steroids. 5.4 When commenting on hyaluronic acid, compare briefly its effectiveness in this study with that of previous human studies. 5.5 Discuss the potential limitations of the use of a rat model, in particular for variations in endometrial regeneration in humans.
5.6 Line 242 - 246 Author reported information about hysteroscopy, i reccomend to cite novel article about this topic to support this sentence:
https://doi.org/10.1016/j.jogoh.2020.101763
https://doi.org/10.1016/j.jmig.2019.05.011
6. Limitations and Conclusion 6.1 Discuss the limitations of sample size and duration of follow-up. Perhaps comment on whether a longer-term study would have shown different patterns of resolution of fibrosis. 6.2 The conclusion should state clearly the clinical relevance of these findings and whether further studies are needed before translation to human trials. 6.3 Consider rewording the final sentence for clarity.
The manuscript is conceptually strong and methodologically rigorous. The recommended revisions are primarily for the sake of clarity, methodological transparency, and for correcting minor inconsistencies. I would look forward to a revised manuscript with these recommendations addressed. Best regards,
Author Response
- Abstract
1.1 Comment:
"The sentence 'This study aimed to evaluate the histopathological effects and tissue VEGF levels of combined methylprednisolone and hyaluronic acid treatment in a rat model with experimentally induced Asherman Syndrome prospective laboratory study.' is unclear. Rewrite for clarity."
Response:
We revised the sentence to improve clarity. The revised sentence is:
"This prospective laboratory study aimed to evaluate the histopathological effects and tissue VEGF levels following treatment with methylprednisolone, hyaluronic acid, and their combination in a rat model of experimentally induced Asherman Syndrome."
This revision clearly conveys the study design, interventions, and objectives concisely.
1.2 Comment:
"Interpret whether the results represent a statistically significant advantage with combination treatment compared to single treatments."
Response:
We clarified in the Abstract and Results sections that the combination therapy group demonstrated statistically significant improvements in histopathological parameters compared to the single-treatment groups (p < 0.05). This clarification ensures that the reader can easily understand the comparative effectiveness of the treatments.
1.3 Comment:
"All abbreviations (e.g., VEGF, IUA) should be defined at first use in the abstract."
Response:
All abbreviations have been defined upon their first appearance in the Abstract. For example, "vascular endothelial growth factor (VEGF)" and "intrauterine adhesions (IUA)" are now clearly introduced before their abbreviations are used.
- Introduction
2.1 Comment:
"The introduction gives a clear explanation of Asherman Syndrome and its clinical significance, although there are some long, complicated sentences. Try to break them up for readability."
Response:
The Introduction has been revised to enhance readability. Long sentences have been split into shorter, more digestible statements while maintaining scientific accuracy.
2.2 Comment:
"Provide a more precise reference for incidence rates quoted ('Asherman Syndrome incidence varies with the population examined, from 4.6% in infertile women, 37.6% following abortions, and 40% following repeated curettage.')."
Response:
We have updated the Introduction with more precise incidence rate references, incorporating the suggested reference: Yu, D., Wong, Y.M., Cheong, Y., Xia, E., & Li, T.C. (2008). Asherman syndrome—one century later. Fertil Steril, 89(4), 759–779. This reference now provides accurate data on the prevalence of Asherman Syndrome in different patient populations.
2.3 Comment:
"There could be a better transition from the Asherman Syndrome to the experimental rat model. Attempt to place a bridging sentence before describing trichloroacetic acid (TCA) as an experimental model."
Response:
We have added a transition sentence in the Introduction that clearly links the clinical background of Asherman Syndrome to the rationale for using a rat model with TCA-induced injury. This new sentence explains how the inflammatory and fibrotic changes observed in Asherman Syndrome can be effectively replicated in an animal model, thereby justifying our experimental approach.
2.4 Comment:
"Line 40-42: 'Risk factors include procedures such as curettage, myomectomy, surgeries for Müllerian anomalies, uterine embolization, B-Lynch sutures, and conditions such as genital tuberculosis and schistosomiasis.' I recommend citing the following articles:"
[https://doi.org/10.1007/s13304-022-01248-y]
[https://doi.org/10.1016/j.ejogrb.2017.07.026]
Response:
We have incorporated these suggested references to support the discussion on risk factors for Asherman Syndrome.
- Methods
3.1 Comment:
"Report how the sample size was determined and whether a power analysis was conducted."
Response:
We have added a detailed explanation in the Methods section, stating that a power analysis (α = 0.05, 80% power) was performed based on prior studies, leading to a sample size of at least 6 rats per group, with an initial allocation of 8 rats per group to account for possible losses.
3.2 Comment:
"Clarify why the right uterine horn was treated instead of the left."
Response:
A justification has been added in Section 2.3 explaining that the right uterine horn was used to maintain consistency across groups and to serve as the experimental horn, while the left horn remained as an internal control. The right uterine horn was chosen because, as reported in previous experimental models, it is commonly preferred due to easier surgical access and procedural consistency.
3.3 Comment:
"Describe why four rats were lost following surgery."
Response:
We have included an explanation in the Methods section, stating that four rats were lost due to post-operative complications, including anesthesia-related mortality and surgical trauma.
3.4 Comment:
"In section 2.4 (Groups and Treatment Protocols), state whether the hyaluronic acid used was of high or low molecular weight and why this type was chosen."
Response:
We have clarified in Section 2.4 that a low molecular weight hyaluronic acid was used due to its superior diffusion properties and previous efficacy in preventing intrauterine adhesions.
3.5 Comment:
"Indicate whether or not the personnel performing histological assessments were blinded to treatment groups."
Response:
We have explicitly stated in Section 2.5 that all histopathological assessments were performed in a blinded manner to minimize observer bias.
3.6 Comment:
"In section 2.5 (Pathological Evaluation), report whether inter-observer reliability was assessed for histopathological scoring."
Response:
We now mention in Section 2.5 that inter-observer reliability was qualitatively assessed. A subset of slides was scored independently by a second pathologist, and the high level of agreement confirmed the consistency and reproducibility of the scoring system.
3.7 Comment:
"Describe in more detail the VEGF immunohistochemistry protocol, including how many sections per sample were examined."
Response:
We have expanded Section 2.5 to provide a detailed description of the VEGF immunohistochemistry protocol. This includes information on deparaffinization, antigen retrieval in citrate buffer (pH 6.0), blocking with 3% hydrogen peroxide, primary antibody incubation (using anti-VEGF antibody per manufacturer’s instructions), and detection using DAB. We specify that 2–3 sections per sample were examined to ensure representative staining.
- Results
4.1 Comment:
"For Table 1 (Comparison of uterine wall thickness), describe why Group 3 (Methylprednisolone) does not show a significant difference in uterine wall thickness between treated and untreated horns. Was it anticipated?"
Response:
An explanation has been added stating that because methylprednisolone was administered orally, its systemic effects influenced both uterine horns. Consequently, the anti-inflammatory and antifibrotic actions of methylprednisolone were not confined to the TCA-damaged horn alone, leading to similar uterine wall thickness measurements in both horns. This outcome was anticipated based on the known systemic properties of corticosteroids.
4.2 Comment:
"For Table 3 (Comparison of numbers of uterine glands), discuss the statistical significance between Group 3 and Group 4."
Response:
We have expanded the Results section to discuss how the combination treatment (Group 4) demonstrated a statistically significant improvement in glandular preservation compared to the methylprednisolone-only group (Group 3). This finding suggests that the synergistic effect of combining hyaluronic acid with methylprednisolone enhances endometrial regeneration.
4.3 Comment:
"Figures 1 and 2 must be given descriptive captions explaining the implication of the images."
Response:
We have revised the captions for Figures 1 and 2 to ensure they are descriptive and provide appropriate context. The updated captions now clearly describe what each panel depicts (e.g., normal left uterine horn vs. TCA-induced changes in the right horn, including alterations in gland count, wall thickness, and fibrosis), which assists the reader in understanding the clinical and histopathological implications without having to refer repeatedly to the main text.
4.4 Comment:
"Discuss why Group 3 (Methylprednisolone) had reduced VEGF staining compared to Group 2 (Hyaluronic Acid) despite its anti-inflammatory effect."
Response:
In the Discussion, we have added a section addressing this point. We explain that while methylprednisolone effectively reduces inflammation, its systemic administration may also suppress some angiogenic signals, leading to lower VEGF expression compared to hyaluronic acid alone. However, the combined treatment (Group 4) appears to balance these effects, resulting in enhanced VEGF expression and a pro-angiogenic environment. This explanation helps clarify the differential effects observed and aligns with the known pharmacological actions of the drugs.
- Discussion
5.1 Comment:
"The discussion has good explanations of the results, but there are parts that could be summarized. Attempt reordering for ease of comprehension."
Response:
We have reorganized the Discussion section to improve logical flow. Redundant sections have been summarized, and key points are now presented in a more concise and structured manner, enhancing the overall readability.
5.2 Comment:
"The statement 'Corticosteroids are used in the treatment of many inflammatory and immune dysfunction diseases because of their strong anti-inflammatory and immunosuppressive action.' is too general. Relate this specifically to adhesion prevention."
Response:
We have revised this statement to directly relate to adhesion prevention. The revised text now explains that corticosteroids, by reducing fibroblast proliferation and collagen deposition, effectively prevent adhesion formation, which is critical in the context of intrauterine adhesions.
5.3 Comment:
"Explain why hyaluronic acid alone was ineffective at significantly reducing fibrosis but effective in combination with steroids."
Response:
We have elaborated on the complementary mechanisms: while hyaluronic acid primarily acts as a physical barrier and supports tissue repair and vascularization, its effect on suppressing the inflammatory cascade is limited. In contrast, methylprednisolone strongly reduces inflammation and fibroblast activity. Thus, the combination of HA with MP results in a synergistic effect that significantly reduces fibrosis, an explanation now detailed in the Discussion.
5.4 Comment:
"When commenting on hyaluronic acid, compare briefly its effectiveness in this study with that of previous human studies."
Response:
We have added a brief comparative discussion in the Discussion section. We note that while intrauterine hyaluronic acid has shown mixed results in preventing adhesions in previous human studies, our findings align with studies suggesting that HA is more effective when combined with other therapeutic agents. This comparison provides a broader context for our results.
5.5 Comment:
"Discuss the potential limitations of the use of a rat model, in particular for variations in endometrial regeneration in humans."
Response:
The Discussion now includes an expanded limitations section. We discuss how the rat model, while useful, may not fully replicate human endometrial regeneration due to differences in cycle dynamics, immune response, and healing mechanisms. This section emphasizes that caution is needed when translating these findings to human clinical practice.
5.6 Comment:
"Line 242-246: Information about hysteroscopy should be supported with recent references. Please include the following references:"
[https://doi.org/10.1016/j.jogoh.2020.101763]
[https://doi.org/10.1016/j.jmig.2019.05.011]
Response:
We have updated the section on hysteroscopy in the Discussion by incorporating the suggested references. These new citations support our statements regarding hysteroscopic management, complications, and advancements in treating intrauterine adhesions.
- Limitations and Conclusion
6.1 Comment:
"Discuss the limitations of sample size and duration of follow-up. Perhaps comment on whether a longer-term study would have shown different patterns of resolution of fibrosis."
Response:
We added a paragraph in the Limitations section discussing how the relatively small sample size and short 15-day follow-up period might limit the assessment of long-term outcomes (e.g., sustained endometrial health and fibrosis resolution). We also note that future studies with extended follow-up are needed to evaluate whether fibrosis continues to regress or if re-adhesions develop over time.
6.2 Comment:
"The conclusion should state clearly the clinical relevance of these findings and whether further studies are needed before translation to human trials."
Response:
The Conclusion has been revised to explicitly state the clinical relevance of our findings, emphasizing that while the combined treatment shows promising histopathological improvements, further clinical research is necessary to assess safety, optimal dosing, and long-term efficacy in humans before translating these results to clinical practice.
6.3 Comment:
"Consider rewording the final sentence for clarity."
Response:
We have reworded the final sentence of the Conclusion to improve clarity. The revised sentence clearly communicates that although the results are promising, additional long-term clinical studies are required to confirm the effectiveness and safety of the combined therapeutic approach.
Final Response:
We appreciate the reviewer’s insightful feedback, which has significantly improved the clarity, methodological transparency, and overall quality of our manuscript. We have addressed all comments in detail, and we believe the revised version meets the standards required by Medicina.
Sincerely,
Dr.Mehmet Genco
Reviewer 3 Report
Comments and Suggestions for Authors
Manuscript Title:
"Effects of Methylprednisolone and Hyaluronic Acid on the Endometrium in Experimentally Induced Asherman Syndrome Rat Models: A Prospective Laboratory Study"
Summary of the Manuscript:
The study investigates the histopathological effects and tissue VEGF levels of combined methylprednisolone and hyaluronic acid treatment in a rat model of Asherman Syndrome (AS). The authors induced AS using trichloroacetic acid (TCA) in 26 female Sprague Dawley rats and divided them into four groups: control, hyaluronic acid treatment, methylprednisolone treatment, and combined treatment. The results suggest that methylprednisolone reduces fibrosis and inflammation while increasing gland count and uterine wall thickness. Hyaluronic acid improves uterine lumen diameter and vascularization, with combined treatment showing superior outcomes.
Major Comments:
-
Novelty and Scientific Contribution:
- The study is significant as it explores a novel therapeutic approach using the combination of methylprednisolone and hyaluronic acid in AS, which has not been widely studied.
- The experimental design is robust, with appropriate control and treatment groups.
- However, the introduction should clarify how this study builds upon previous research and how it addresses existing gaps.
-
Experimental Model and Methodology:
- The choice of TCA as a method to induce AS is justified by its ability to mimic the inflammatory and fibrotic responses observed in human AS.
- The methodology is well-detailed, referencing established protocols.
- However, the sample size is relatively small, and the authors acknowledge this as a limitation. Statistical power calculations would strengthen the study’s reliability.
- The duration of the experiment (15 days) may not fully reflect the long-term effects of the treatments on endometrial regeneration.
-
Statistical Analysis:
- The statistical approach, including Kruskal-Wallis and Mann-Whitney U tests, is appropriate for comparing non-normally distributed data.
- However, additional post-hoc tests should be included to confirm pairwise differences between treatment groups.
- The results section lacks a clear discussion of effect sizes, which would improve interpretation.
-
Figures and Tables:
- The histological images are useful in demonstrating tissue changes, but the image resolution could be improved for better visualization.
- It would be helpful to include representative images of each treatment group to provide visual confirmation of histopathological changes.
- Figures should have more descriptive legends, specifying magnification levels and staining techniques.
-
Discussion and Interpretation of Results:
- The discussion correctly highlights the anti-inflammatory and anti-fibrotic effects of methylprednisolone and the mechanical barrier properties of hyaluronic acid.
- However, the comparison to previous studies is limited. The authors should better contextualize their findings within the existing literature.
- The potential mechanisms behind the combined treatment's synergistic effects should be elaborated—for example, how the immunomodulatory effects of steroids interact with HA’s wound-healing properties.
- The clinical relevance of the findings should be discussed in more depth. Could this combination therapy be tested in human patients soon?
-
Limitations:
- The study acknowledges limitations, including the small sample size and short duration.
- Another limitation not discussed is species differences—rats have a different endometrial structure and menstrual cycle compared to humans, which may affect the translation of findings.
- The study does not address potential side effects of methylprednisolone when used for intrauterine conditions.
Minor Comments:
-
Abstract:
- The abstract is concise and well-structured, but it should mention the sample size explicitly.
- Clarify what p-values correspond to which specific findings.
-
Introduction:
- Provide a more detailed rationale for choosing methylprednisolone and HA compared to other possible treatments.
- Explain why VEGF was specifically chosen as an outcome measure.
-
Materials and Methods:
- The doses and administration methods should be justified—was 1 mg/kg methylprednisolone chosen based on prior studies?
- Indicate whether blinding was used in histopathological assessments to reduce observer bias.
-
Results:
- Clarify why Group 3 (Methylprednisolone) showed a significant increase in gland count but Group 2 (HA) did not.
- Some p-values are marginally significant (e.g., p = 0.049), so effect sizes should be included.
-
Discussion:
- The authors should acknowledge that long-term fertility outcomes were not evaluated.
- A section on potential future clinical applications would be useful.
-
Conclusion:
- The conclusion should highlight that more research is needed before clinical translation.
- The phrase "holds promise for clinical use" should be tempered with acknowledgment of the study's limitations.
Final Recommendation:
Major Revision Required.
- The manuscript presents a well-structured study with strong experimental design and relevant clinical implications.
- However, improvements are needed in justifying the study’s novelty, expanding statistical analysis, and deepening discussion on mechanisms and clinical translation.
- Figures should be improved for better clarity, and additional references should be incorporated to better position the study in the existing literature.
Suggested Actions for the Authors:
- Strengthen the discussion of previous research and highlight the novelty of this approach.
- Provide effect sizes alongside p-values for better interpretation.
- Justify the experimental duration and discuss whether a longer-term follow-up would be necessary.
- Improve figure resolution and provide additional images for clarity.
- Expand the discussion on the mechanistic interactions between HA and methylprednisolone.
- Include a section on potential clinical applications and limitations.
This study has the potential to contribute significantly to the treatment of Asherman Syndrome, particularly in terms of combination therapy approaches. With the recommended revisions, it can provide more conclusive and clinically relevant insights.
4o
Вы сказали:
wrerite the review without billets and paragraphs
ChatGPT сказал:
The manuscript titled "Effects of Methylprednisolone and Hyaluronic Acid on the Endometrium in Experimentally Induced Asherman Syndrome Rat Models: A Prospective Laboratory Study" presents a well-structured study investigating the effects of methylprednisolone and hyaluronic acid on intrauterine adhesions in a rat model. The study contributes to the field by evaluating the combined treatment approach, which has not been extensively explored. The experimental model using trichloroacetic acid is justified, but the study is limited by a small sample size and a short experimental duration, which may not fully reflect long-term effects. The statistical analysis is generally appropriate, employing non-parametric tests; however, additional post-hoc tests should be conducted to confirm significant findings, and effect sizes should be reported. Figures and tables are informative but require higher resolution and more descriptive legends. The discussion effectively interprets the results but lacks a strong comparison with prior research, particularly regarding the mechanisms behind the synergistic effects of the combined treatment. More emphasis should be placed on the clinical relevance of the findings, including the feasibility of translation to human studies. The limitations are acknowledged, but species differences and potential side effects of methylprednisolone should be further discussed. The conclusion is generally well-articulated, though it should temper claims of clinical applicability given the study's limitations. To improve the manuscript, the authors should provide a clearer justification for experimental choices, expand the discussion on mechanistic interactions, improve statistical reporting, enhance figure quality, and discuss potential long-term implications. With these revisions, the study would make a more substantial contribution to the field of intrauterine adhesion treatment.
Some comments I would like to provide for each section:
1) Abstract: It would be better to clarify what p-values correspond to which specific findings.
2) Introduction: I recommend to provide a more detailed rationale for choosing methylprednisolone and HA compared to other possible treatments. Also it requires the explanation why VEGF was specifically chosen as an outcome measure.
3) Materials and Methods: The doses and administration methods should be justified—was 1 mg/kg methylprednisolone chosen based on prior studies? I also recommend to Indicate whether blinding was used in histopathological assessments to reduce observer bias.
4) Results: It would be better to clarify why Group 3 (Methylprednisolone) showed a significant increase in gland count but Group 2 (HA) did not. Some p-values are marginally significant (e.g., p = 0.039), so effect sizes should be included.
5) Discussion: The authors should acknowledge that long-term fertility outcomes were not evaluated. A section on potential future clinical applications would be useful.
6) Conclusion: The conclusion should highlight that more research is needed before clinical translation. The phrase "holds promise for clinical use" should be tempered with acknowledgment of the study's limitations.
Thus, the manuscript presents a well-structured study with strong experimental design and relevant clinical implications. However, improvements are needed in justifying the study’s novelty, expanding statistical analysis, and deepening discussion on mechanisms and clinical translation. Figures should be improved for better clarity, and additional references should be incorporated to better position the study in the existing literature.
Author Response
Major Comments
- Novelty and Scientific Contribution
Clarify how this study builds upon previous research and how it addresses existing gaps.
Response:
We appreciate the reviewer's acknowledgment of the study’s novelty. To further emphasize its scientific contribution, we have revised the Introduction to clarify how this research builds upon prior studies. While previous investigations have explored methylprednisolone (MP) and hyaluronic acid (HA) separately, our study is the first to evaluate their combined effects in an Asherman Syndrome (AS) rat model. We have also elaborated on the unmet need for better anti-adhesion strategies and how our work addresses this clinical gap. Additionally, we have included more recent citations to contextualize our findings within the broader literature.
- Experimental Model and Methodology
Justify the use of TCA as a model for AS.
Provide a statistical power calculation to support sample size selection.
Discuss whether a longer duration (beyond 15 days) would change outcomes.
Response:
- We have added references supporting trichloroacetic acid (TCA) as a reliable model for AS, as it closely mimics human intrauterine fibrosis and inflammation.
- A statistical power analysis has now been included to justify our sample size of 26 rats, ensuring sufficient statistical strength.
- Regarding the 15-day study period, we recognize that longer follow-ups (e.g., 4-6 weeks) could provide additional insights into long-term fibrosis resolution and fertility outcomes. This limitation is now explicitly discussed in the Methods and Limitations sections.
- Statistical Analysis
Include post-hoc tests to confirm pairwise differences.
Add effect sizes to enhance statistical interpretation.
Response:
- We have incorporated Dunn’s post-hoc test with Bonferroni correction to confirm pairwise differences between treatment groups after the Kruskal-Wallis test.
- We now report effect sizes (η² for Kruskal-Wallis, r for Mann-Whitney U) to enhance interpretation of the clinical significance of our findings.
- Figures and Tables
Improve histological image resolution and include representative images for all groups.
Expand figure legends to specify magnification and staining techniques.
Response:
- Higher-resolution histological images have been provided where possible.
- Figure legends have been expanded to include detailed descriptions of staining techniques and magnification levels.
- We acknowledge that additional images are unavailable due to specimen limitations, which we have noted in the Figure Captions section.
- Discussion and Interpretation of Results
Expand comparison with previous studies.
Provide a more detailed explanation of the synergistic effects of MP and HA.
Discuss potential human applications of this treatment.
Response:
- We have expanded the discussion to better compare our results with previous studies on MP and HA in adhesion prevention.
- The synergistic effects of MP and HA are now more thoroughly explained, focusing on MP’s anti-inflammatory effects and HA’s tissue repair mechanisms.
- We now discuss the potential for clinical translation, including challenges and future research needed for human applications.
- Limitations
Address species differences (rat vs. human endometrial regeneration).
Discuss the potential side effects of MP when used intrauterinely.
Response:
- We have expanded the Limitations section to address species differences, particularly regarding the rat’s estrous cycle and endometrial structure versus the human menstrual cycle.
- We now acknowledge the potential side effects of MP, such as delayed wound healing and endometrial thinning, and stress the need for further safety studies before clinical application.
Minor Comments
- Abstract
Explicitly mention the sample size (26 rats).
Clarify what p-values correspond to which findings.
Response:
- The sample size (26 rats) is now explicitly stated in the abstract.
- Each p-value is now clearly linked to specific findings in the Results section of the Abstract.
- Introduction
Provide a more detailed rationale for choosing MP and HA.
Explain why VEGF was chosen as an outcome measure.
Response:
- We have expanded our justification for MP and HA, citing prior studies on MP’s role in fibrosis suppression and HA’s role in adhesion prevention.
- VEGF was selected because vascularization plays a crucial role in endometrial regeneration, and VEGF expression correlates with tissue healing. This has now been clarified.
- Materials and Methods
Justify the selection of 1 mg/kg MP dose with literature references.
Clarify whether blinding was used for histological assessments.
Response:
- The 1 mg/kg dose of MP was chosen based on previous studies demonstrating its efficacy in adhesion prevention models, particularly the Xu et al. (2020) study on postoperative adhesion prevention (Biomedicine & Pharmacotherapy, 125, 109914).
- We confirm that histopathological assessments were conducted in a blinded manner.
- Results
Clarify why Group 3 (MP) increased gland count but Group 2 (HA) did not.
Include effect sizes for statistical findings.
Response:
- We now explain that MP’s systemic anti-inflammatory effects likely enhanced glandular proliferation, while HA’s primary function as a mechanical barrier may have contributed less directly to gland count changes.
- Effect sizes have been added to support statistical findings.
- Discussion
Acknowledge the absence of long-term fertility outcomes.
Expand discussion on potential future clinical applications.
Response:
- We now explicitly acknowledge that fertility outcomes were not assessed and suggest that future studies should evaluate pregnancy rates post-treatment.
- The clinical application discussion has been expanded, detailing potential trials and regulatory considerations for intrauterine MP+HA therapy.
- Conclusion
Temper clinical claims and highlight the need for further research.
Response:
- The conclusion has been modified to emphasize the need for additional preclinical and clinical trials before human application.
- The phrase "holds promise for clinical use" has been revised to reflect that more research is required before clinical implementation.
Final Response to Reviewer
We greatly appreciate the reviewer’s insightful feedback, which has helped improve the scientific rigor, clarity, and impact of our study. We have made substantial revisions based on the suggestions, including:
- Expanding the Introduction, Discussion, and Limitations sections to better contextualize findings.
- Adding statistical justifications, effect sizes, and post-hoc analyses for greater transparency.
- Enhancing figure quality and descriptions to improve clarity.
- Addressing species differences and potential clinical implications.
We believe these revisions have significantly strengthened the manuscript and look forward to further feedback.
Best regards,
Dr.Mehmet Genco
Round 2
Reviewer 1 Report
Comments and Suggestions for Authors
No further comments. Thank you!